# *Pelargonium graveolens*: Towards In-Depth Metabolite Profiling, Antioxidant and Enzyme-Inhibitory Potential

**DOI:** 10.3390/plants13182612

**Published:** 2024-09-19

**Authors:** Reneta Gevrenova, Gokhan Zengin, Vessela Balabanova, Anna Szakiel, Dimitrina Zheleva-Dimitrova

**Affiliations:** 1Department of Pharmacognosy, Faculty of Pharmacy, Medical University-Sofia, 2 Dunav Str., 1000 Sofia, Bulgaria; vbalabanova@pharmfac.mu-sofia.bg (V.B.); dzheleva@pharmfac.mu-sofia.bg (D.Z.-D.); 2Department of Biology, Science Faculty, Selcuk University, Konya 42130, Turkey; gokhanzengin@selcuk.edu.tr; 3Department of Plant Biochemistry, Faculty of Biology, University of Warsaw, 1 Miecznikowa Street, 02-096 11 Warsaw, Poland; a.szakiel@uw.edu.pl

**Keywords:** *Pelargonium graveolens*, phenolics, antioxidants, enzyme inhibition

## Abstract

*Pelargonium graveolens* L’Hèr. (Geraniaceae) is renowned for its traditional use as a flavor, ornamental and medicinal plant. This work aimed at an in-depth study of the phytochemical profiling and in vitro antioxidant and enzyme inhibition assessment of a methanol-aqueous extract from *P. graveolens* leaves. A UHPLC-HRMS analysis revealed more than 110 secondary metabolites, including 8 acyltartaric and 11 acylcitric/acylisocitric acids; 8 gallotannins; 36 flavonols, flavanones and methoxylated flavonoids together with 17 phenolic and aliphatic acids; and 21 phenolic acid glycosides. For the first time, acylcitric acids along with feruloyl- and coumaroyltartaric acids are reported in the species. The leaf extract actively scavenged 2,2-diphenyl-1-picrylhydrazyl DPPH (273.45 mg trolox equivalent (TE/g)) and 2,2′-azino-bis(3-ethylbenzothiazoline-6-sulfonic acid) (ABTS^•+^) radicals (531.97 mgTE/g) and showed a high reducing power: 431.32 mg TE/g Cupric reducing antioxidant capacity (CUPRAC) and 292.21 mg TE/g Ferric reducing antioxidant power (FRAP). It possessed a metal chelating capacity (13.44 ethylenediaminetetraacetic acid equivalent (EDTAE)/g) and contained 2.71 mmol TE/g in the phosphomolybdenum assay. The rose geranium extract exhibited high inhibition towards acetyl- and butyrylcholinesterase (2.80 and 2.20 mg galantamine equivalent (GALAE)/g, respectively) and tyrosinase (75.49 mg kojic acid equivalent (KAE)/g). It inhibited α-glucosidase and α-amylase (3.75 mmol and 0.79 acarbose equivalent (ACAE)/g, respectively) and lipase (28.91 mg orlistat equivalent (OE)/g). This study sheds light into the future potential application of the rose geranium in pharmaceutical and nutraceutical products.

## 1. Introduction

*Pelargonium graveolens* L’Hèr. (Geraniaceae) is renowned for its traditional use as a flavor, ornamental and medicinal plant [1,2]. The species is native to Southern Africa but has also been widely naturalized in Europe, North America, the Middle East, Eastern Africa and Australia [3]. *P. graveolens* is commonly referred to as rose geranium and rose-scented geranium. It is an erect, branched shrub with herbaceous stems becoming woody at the base, up to 1.3 m high [1,4]. The plant is of economic importance [5,6]. It is cultivated throughout the whole world for its traditional usage to flavor food and beverages, tea, potpourris and perfumes [7]. As the essential oil is highly evaluated, attempts have been made to promote its cultivation by employing various agro- and biotechnology approaches [3,8,9]. There are different *P. graveolens* cultivars, such as CIMAP Bio-G-171 and CIM-Bharat, which are difficult to be differentiated, and the composition of their oil varies significantly [10,11,12]. Fresh and dried rose geranium leaves have a fine rosy odor with a pronounced fruity minty undertone and a sweet astringent taste. Rose geranium essential oil (EO) is secreted in the glandular trichomes. EO accounted for 0.11-0.49% from the leaves [5]. Overall, more than 200 compounds have been reported in the essential oils [1,6]. High levels of oxygenated monoterpenes have been evidenced, with an average concentration ranging from 64.3 to 74.2% [6,8]. Geraniol (up to 43%), citronellol (up to 33%), linalool (up to 50%) and their esters dominate the EO [5]. Non-oxygenated sesquiterpenes are the most abundant within the sesquiterpenoids—δ-selinene, *β*-caryophyllene and guaia-6, and 9-diene reach up to 8.15, 7.7 and 6.58%, respectively [1]. According to their main constituents, three types of rose geranium essential oils have been distinguished: the Chinese type, which contains a great amount of citronellol (30–40%); the African type, originating from Algeria, Morocco and Egypt, which is characterized by 10-epi-γ-eudesmol (4–5%); and the Bourbon type from Reunion Island or Madagascar, which is delineated by guaia-6, 9-diene (5–7%), geraniol (15–18%) and linalool (0.5–8%) [5]. Rose geranium contains phenolic acids, flavanols, flavanones, flavan-3-ols, tannins and coumarins [1]. Few quercetin, kaempferol and myricetin glycosides have been putatively identified by HPLC–PDA–ESI–MS/MS and UPLC-ESI-MS [7,13]. Androutsopoulou et al. [7] reported several flavonoids in the ethanol-aqueous extracts from the leaves. Quercetin 3-*O*-glucoside dominated the flavonoids at 17.7 μg/mL, while Kaempferol *O*-pentosyl-*O*-glucuronide and Quercetin 3-*O*-galactoside reached 15.7 and 13.9 μg/mL, respectively.

Rose geranium leaves have been used in an ethnopharmacological approach as a wounds healing, astringent, anti-diabetic, immune stimulant and pain relief agent [5]. In Africa, the main use is to treat gastrointestinal disorders, respiratory infections and skin diseases [1]. The traditional use, phytochemistry and pharmacology of rose geranium were the subject of a literature review by Amel et al. [1], which particularly emphasized the anticancer and anti-diabetic potential of the herbal drug. Roman et al. [5] reviewed the progress of the pharmacological activity, volatile compounds and perspectives of *Pelargonium* sp. essential oils and highlighted that this scented plant should be rediscovered for a sustainable exploitation in the food processing industry and nutraceuticals. *P. graveolens* EO has been used for many years in traditional medicine as antiasthmatic, antiallergic, antidiarrhoeic and antidiabetic agent [3,14,15]. The anti-inflammatory, antimicrobial and antioxidant activity of the EO and extracts sustain the traditional claims relating to rose geranium’s use for mitigation of inflammatory issues [7,16]. Significant advances have been made in understanding the mechanisms underlying the EO-mediated anti-inflammatory effect, which is related to the inhibition of the release or synthesis of some mediators (histamine, serotonin, prostaglandins, nitric oxide) [16]. In addition to evoking an antioxidant response, rose geranium EO displays anti-diabetic and hypolipidemic activity in animal models, which generates further interest in the herbal drug as a potential candidate for metabolic chronic diseases [14,16]. 

It is worth noting that the aforementioned studies emphasized the essential oil’s composition and its health promoting effects, while there is no comprehensive metabolite profile of rose geranium leaves by means of liquid chromatography—Orbitrap—high-resolution mass spectrometry (LC-HRMS). 

The metabolic study of medicinal plants aims to provide a comprehensive examination of metabolite profiles and quality assessment of herbal drugs [17]. Owing to the technological progress of hyphenated analytical methods, the comprehensive metabolite profiling of medicinal plants has seen an exponential increase [18]. Essential aspects in plant metabolite profiling have been recently emphasized in the valuable study by Çiçek et al. [18]. 

In view of these studies, we undertook an in-depth profiling of the secondary metabolites of methanol-aqueous extract from *P. roseum* leaves by means of ultra-high-performance liquid chromatography—Orbitrap high-resolution mass spectrometry, integrated with an assessment of its antioxidant and enzyme-inhibitory potential. 

## 2. Results and Discussion

### 2.1. Dereplication and Annotation of Specialized Metabolites in Pelargonium roseum Extracts

In order to be of maximum scientific relevance, the *P. graveolens* metabolite profiling was consistent with the approach of Çiçek et al. [18], including (i) a critical review of the previous literature and a rational sampling strategy; (ii) transparent plant sampling with cultivated material documented by vouchers in public herbaria (iii); (iv) transparent, documented state-of-the art chemical analysis, including chemical reference standards; (v) use of gentle analytical methods; (vi) careful chemical data interpretation, avoiding over- and misinterpretation and taking into account phytochemical complexity when assigning identification confidence levels; and (vii) taking all previous scientific knowledge into account in reporting the scientific data [18]. A complete workflow revealing the essential aspects of qualitative plant metabolite profiling is presented in Figure 1. The total ion chromatogram of the assayed extract is depicted in Appendix A. MS/MS fragmentation patterns of the annotated/dereplicated compounds are described in Appendix A.

Identification confidence levels for *P. graveolens* metabolite profiling were performed according to Çiçek et al. [18] and were as follows: 

**A2**: Confirmed structure including confirmed stereochemistry;

**B**: Confirmed structure except for one or more stereochemical aspects; 

**C:** Tentative identification matched with a standard compound, match of at least tR, MS and MS/MS with an actual authentic standard analyzed in parallel, preferably supported by other online data; 

**D:** Tentative identification based on libraries, model compounds, etc.; **D1**: relatively reliable evidence; **D2:** relatively poor evidence; 

**E:** Tentative candidate or tentative identification of metabolite class [18].

#### 2.1.1. Phenolic Acids and Their Glycosides, Coumarins and Aliphatic Acids

Based on the accurate MS masses and conformity of the fragmentation patterns and retention times of reference standards, six hydroxybenzoic acids (**5**, **9**, **20**, **21**, **36**, **43** and **47**) and five hydroxycinnamic acids (**24**, **29**, **34**, **35** and **40**) were identified in the assayed extract (Table 1, Appendix A and Appendix A). Their recognition was founded on the diagnostic fragment ions reported elsewhere [19,20]. Additionally, nine hydroxybenzoic acid hexosides (**6**–**8**, **10**–**13**, **22** and **30**) and seven hydroxycinnamic acid hexosides (**14**–**16**, **23**, **26**, **28** and **41**) together with two sugar esters (**13** and **19**) were annotated. Among them, four caffeic and dihydrocaffeic acid conjugates (**14**, **16**, **23** and **26**) were established.

Compounds **39** and **42** shared the same [M-H]^−^ at *m*/*z* 461.167, which is a formic acid adduct of the deprotonated molecule at *m*/*z* 415.161 (**39** consistent with C_19_H_27_O_10_, 0.457 ppm) (Table 1). A series of prominent fragment ions at *m*/*z* 311.099 [M-H-C_8_H_8_]^−^, 293.087 [M-H-C_8_H_8_O]^−^ accompanied by the hexose cross cleavages at *m*/*z* 251.077 [M-H-C_8_H_8_-^0,4^Hex]^−^, 221.066 [M-H-C_8_H_8_-^0,3^Hex]^−^ and 191.055 [M-H-C_8_H_8_-^0,2^Hex]^−^ suggested phenylethyl alcohol (C_8_H_8_O, 122 Da) and a hexose unit. A disaccharide (pentosylhexoside) was deduced from the fragment ions at *m*/*z* 179.055 [M-H-C_8_H_8_-Pent]^−^ and 161.045 [M-H-C_8_H_8_-(Pent+H_2_O)]^−^. Consequently, **39** and **42** were related to phenylethyl alcohol-pentosylhexoside isomers. In addition to the commonly found simple coumarins esculetin *O*-hexoside (**18**), esculetin (**32**) and scopoletin *O*-hexoside (**25**, **33**), caffeoyl ester of scopoletin-*C*-hexoside (**44**) was evidenced by the series of ions resulting from hexose cross-cleavages, as has been seen in the *C*-glycosides [20]. Scopoletin and caffeoyl moiety were deduced from the fragment ions at *m*/*z* 191.034 [Scopoletin-H]^−^, 161.024 [Caffeic acid-H]^−^ and 135.044 [Caffeic acid-H-CO_2_]^−^. Concerning aliphatic acids, two tricarboxylic acid isomers (citric and isocitric acid), together with the dicarboxylic tartaric and malic acids, were annotated in the studied extract. Additionally, **45** afforded a precursor ion at *m*/*z* 403.160 (C_18_H_27_O_10_) together with distinctive fragments at *m*/*z* 241.108 [M-H-Hex]^−^ and 223.097 [Glansreginic acid-H]^−^ (base peak), along with 197.118 [Glansreginic acid-H-CO_2_]^−^ and 89.023 [Glansreginic acid-H-CO_2_-C_8_H_12_]^−^. Thus, **45** was ascribed to Glansreginic acid *O*-hexoside [21]. 

MS/MS spectrum of **48** was acquired. The prominent ions at *m*/*z* 337.093 [M-H-168.042]^−^, 193.049 [Ferulic acid-H]^−^ and 175.039 [Ferulic acid-H-H_2_O]^−^ suggested the subsequent losses of vanillic acid (168 Da) and hexose moiety (162 Da) supported by the base peak at *m*/*z* 167.034 [Vanillic acid-H]^−^ and abundant fragment ions at *m*/*z* 160.015 [Ferulic acid-H-H_2_O-CH_3_]^−^ (31.9%) and 152.010 [Vanillic acid-H-CH_3_]^−^ (63%). The lack of the hexose cross-cleavage indicated glycosidic bonds in the molecule. Accordingly, **48** was tentatively identified as Ferulic acid-4-*O*-(vanillyl)-hexoside. Gentisic acid-*O*-(feruloyl)-hexoside (**49**) was deduced from the following transitions: 491.120 → 315.072 [M-H-C_10_H_8_O_3_]^−^ and 315.072 → 153.018 [(M-H-C_10_H_8_O_3_-C_6_H_10_O_6_)]^−^ corroborated by the fragment ion at *m*/*z* 193.0500 [Ferulic acid-H]^−^. Within this group, the profile was dominated by the following compounds: tartaric acid (**1**) (25.36%), mallic acid (**2**) (12.29%), citric/isocitric acid (**3**) (5.95%), citric/isocitric acid (**4**) (4.69%) and protocatechuic acid *O*-hexoside (**7**) (4.37%). Overall, the phenolic and aliphatic acids and derivatives accounted for 65.55% of the assayed compounds. 

#### 2.1.2. Acyltartaric Acids

In the fragmentation pattern of acyltartaric acids, the deprotonated molecule of tartaric acid (TA) at *m*/*z* 149.008 (C_6_H_11_O_7_) was commonly found, supported by the fragment ions at *m*/*z* 112.987 [TA-H-2H_2_O]^−^, 103.002 [TA-H-H_2_O-CO]^−^, 87.007 [TA-H-H_2_O-CO_2_]^−^ and 72.992 [TA-H-H_2_O-CO-CH_2_O]^−^ (Table 1, Appendix A and Appendix A) [22]. The assignment of two isobars of caffeoyltartaric (caftaric) acid (**50** and **52**) (at *m*/*z* 311.042) was confirmed by the fragment ions at *m*/*z* 179.038 [caffeic acid-H]^−^, 161.023 [caffeic acid-H-H_2_O]^−^ and 135.044 [caffeic acid-H-CO_2_]^−^. Compounds **56** and **57** were consistent with coumaroyltartaric and feruloyltartaric acid affording a base peak at *m*/*z* 163.039 [coumaric acid-H]^−^ and 193.050 [ferulic acid-H]^−^, respectively (Figure 2A). In the same manner, caffeoyltartaric acid-hexoside (**51**) was deduced from the fragment ions at *m*/*z* 341.088 [M-H-C_4_H_4_O_5_]^−^ and 179.034 [M-H-C_4_H_4_O_5_-Hex]^−^, resulting from the subsequent losses of tartaric acid (132 Da) and hexose (162 Da). In addition, caffeoyltartaric acid dimer (**53**) was also observed, as has been previously reported for caftaric acid [23]. Thus, the mass spectrum at 2.36 min consisted of two ions at *m*/*z* 311.041 and 623.089; the former was a deprotonated caffeoyltartaric acid, while the latter was a neutral caftaric acid bond to the deprotonated caftaric acid molecule. In the same way, compounds **54** and **55** (at *m*/*z* 491.083, C_22_H_19_O_13_) were ascribed to dicaffeoyltartaric acid corresponding to a neutral caffeic acid bond to a deprotonated caffeoyltartaric acid molecule. Herein, except for the caftaric acid, acyltartaric acids were reported for the first time in *P. graveolens*. It worth noting that acyltartaric acids reached up to 18.16% of the annotated compounds with a prevalence of caftaric acid isomer 1 and 2 (2.37% and 9.75%, respectively). 

#### 2.1.3. Acylcitric/Acylisocitric Acids

A variety of acylcitric/isocitric acids (AC/IA), including six caffeoyl C/IA (**58**-**62**, **64**), three coumaroyl C/IA (**63**, **65**, **66**) and two feruloyl C/IA (**67**, **68**), were found in the assayed extract (Table 1, Appendix A and Appendix A). Generally, the abundant fragment ion at *m*/*z* 191.019 pointed out the deprotonated molecule of citric acid or its isomer isocitric acid, accompanied by the fragment ions at *m*/*z* 147.029 [CA/IA-H-CO_2_]^−^, 103.002 [CA/IA-H-C_3_H_4_O_3_]^−^ and 85.028 [CA/IA-H-H_2_O-2 CO_2_]^−^. The assignment of caffeoyl esters was based on the fragment ion at *m*/*z* 179.034 [caffeic acid-H]^−^ and corresponding fragments (see caffeoyltartaric acid), while coumaroyl esters were evidenced by the fragment ions at *m*/*z* 163.039 [coumaric acid-H]^−^ and 119.049 [coumaric acid-H-CO_2_]^−^ (Figure 2B). Feruloyl esters were deduced from the prominent fragment ions at *m*/*z* 193.050 [ferulic acid-H]^−^, 134.036 [ferulic acid-CO_2_-CH_3_]^−^ (Appendix A). Herein, citric/isocitric acid derivatives in the *Pelargonium* genus are reported for the first time. They accounted for 2.30% of the assayed secondary metabolites in the profiling.

**Figure 2 plants-13-02612-f002:**
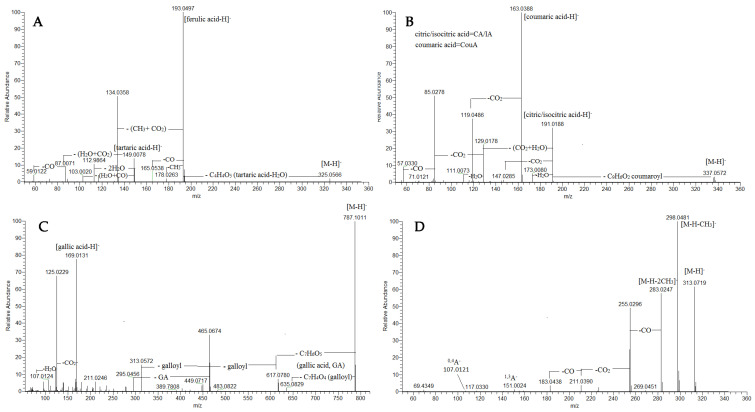
MS/MS spectra of feruloyltartaric acid (**57**) (**A**), coumaroylcitric/coumaroylisocitric acid (**63**) (**B**), tetragalloyl hexoside (**75**) (**C**) and kaempferol dimethyl ether (**111**) (**D**).

#### 2.1.4. Gallotannins

MS/MS fragmentation pattern of gallic acid (GA) (**5**) and its derivatives afforded characteristic fragment ions at *m*/*z* 169.013 [GA-H]^−^, 151.002 [GA-H-H_2_O]^−^, 125.023 [GA-H-CO_2_]^−^ and 107.012 [GA-H-H_2_O-CO_2_]^−^ (Table 1, Appendix A, Appendix A). Two compounds, **69** and **70**, were assigned as gallic acid *O*-hexoside. By analogy with vanillyl-hexose (**13**) and syringoyl-hexose (**19**), **71** was ascribed to the sugar ester digalloyl *O*-hexose, where hexose cross-cleavages were registered at *m*/*z* 271.046 (−60 Da, ^0,4^Hex) and 211.024 (−120 Da, ^0,2^Hex).

MS/MS spectrum of **72** with [M-H]^−^ at *m*/*z* 305.067 was acquired (Table 1). The prominent fragment ions at *m*/*z* 261.077 resulted from the CO_2_ loss of the ring-A cleavage and subsequent ethenone (C_2_H_2_O) loss at *m*/*z* 219.066. The neutral ring-B loss gave rise to an abundant ion at *m*/*z* 179.034 (27.6%), while the base peak at *m*/*z* 125.021 referred to ring-A of the flavanolic skeleton after heterocycle ring cleavage. Accordingly, **71** was annotated as gallocatechin [24].

Concerning **75**, four galloyl residues were evidenced by the transitions at *m*/*z* 787.101 → 635.089 [M-H-galloyl]^−^ → 313.057 [M-H-3galloyl-H_2_O]^−^ together with the prominent ions at *m*/*z* 169.013 and 151.002 (Table 1, Appendix A, Figure 2C). Accordingly, 75 was annotated as tetragalloyl-hexoside [25]. Methylgallate (**74**) with [M-H]^−^ at *m*/*z* 183.029 was deduced from the typical loss of methyl radical (•CH_3_) (−15 Da) at *m*/*z* 168.005 (**9**) (Table 1, Appendix A).

The MS/MS fragmentation pathway of ellagic acid (EA) (**76**) yielded characteristic ions resulting from the neutral losses of CO and CO_2_ at *m*/*z* 257.009 [EA-H-CO_2_]^−^, 229.014 [EA-H-CO-CO_2_]^−^, 201.018 [EA-H-2CO-CO_2_]^−^, 185.023 [EA-H-2CO-2CO_2_]^−^, 173.023 [EA-H-3CO-CO_2_]^−^, 145.028 [EA-H-4CO-CO_2_]^−^ and 129.033 [EA-H-3CO-2CO_2_]^−^(Table 1, Appendix A). Thus, **76** was unambiguously identified by comparison with the reference standard. Overall, the annotated metabolites in the frame of the group reached up to 1.82% of the assayed compounds.

#### 2.1.5. Flavonoids

Flavonoids belong to the class of flavanols and flavanones. They mainly possess kaempferol, quercetin and myricetin as an aglycone moiety in mono- and diglycosides (Table 1, Appendix A). The strategy for flavonoid recognition was based on the fragmentation patterns and diagnostic ions for different classes of flavonoids and flavonoid references [26,27].

A series of flavonoids were closely associated with the same fragmentation pattern giving characteristic fragment ions at *m*/*z* 287.056 (**77**), 301.035 (**84**, **87**), 331.045 (**88**), 273.077 (**93**), 285.040 (**95**), 315.045 (**96**), 271.061 (**97**) and 313.072 (**105**) [(M-H)-Hex]^−^, indicating hexosides (Table 1, Appendix A and Appendix A). In addition to the aglycone Y_0_^−^, an abundant radical aglycone [Y_0_-H]^−•^ was also yielded (quercetin, kaempferol and myricetin), suggesting 3-*O*-hexosides [28]. Based on the comparison with the retention times and fragmentation patterns of reference standards, **83**, **84**, **87**, **95**, **96** and **97** were unambiguously identified as myricitrin, hyperoside, isoquercitrin, astragalin, isorhamnetin 3-*O*-glucoside and naringenin 7-*O*-glucoside, respectively. Compound **101** was assigned to caffeoyl ester of quercetin*O*-hexoside (Table 1, Appendix A). The losses of 162.030 Da (C_9_H_6_O_3_) and 324 Da (C_15_H_16_O_8_) in the fragmentation pattern together with the fragment ions at *m*/*z* 179.034 [caffeic acid-H]^−^, 161.023 [(caffeic acid-H)-H_2_O]^−^and 135.044 [(caffeic acid-H)-CO_2_]^−^ corroborated caffeoyl moiety. 

In particular, the precursor ion of **93** at *m*/*z* 435.130 underwent the loss of 90 and 120 Da according to the sugar cross-ring cleavages ^0,3^Hex and ^0,2^Hex, respectively, suggesting a *C*-glycosidic bond. On the other hand, the aglycone at *m*/*z* 273.077 (C_13_H_15_O_5_) was discernable by the fragment ions at *m*/*z* 167.034 and 125.023, both originating from the ring-A cleavage, as has been observed in the dihydrochalcone phloretin [29]. Accordingly, **93** was assigned as phloretin *C*-hexoside. In the same way, **87** was annotated as phloretin di*C*-hexoside. Prominent fragment ions at *m*/*z* 387.109 [M-H-(120 + 90)]^−^ and 357.098 [M-H-2 × 120]^−^ suggested two *C*-bonded hexose units [20]. Compounds **79**, **82**, **92** and **94** were assigned as rutinosides of myricetin, quercetin, kaempferol and isorhamnetin, respectively. The commonly found loss of 308 Da in the fragmentation patterns of the aforementioned compounds corroborate with the presence of the rutinose, additionally confirmed by the comparison with reference standards. 3-*O*-pentosylhexosides of myricetin (**78**), quercetin (**80**) and kaempferol (**86**), together with their pentosides (**81**, **91** and **98**), were also annotated in the rose geranium extract. 

The ESI-MS/MS spectra of a series of the methyl ether of quercetin, kaempferol and myricetin were acquired (Table 1). Compounds **104** at *m*/*z* 345.062, **109** at *m*/*z* 329.067 and **111** at *m*/*z* 313.072 showed fragment ions by two methyl radical losses at *m*/*z* 315.015 (**104**), *m*/*z* 299.020 (**109**) and 283.025 (**111**) along with RDA ions at *m*/*z* 151.002 and 107.012 (A-ring cleavages). Thus, myricetin-, quercetin and kaemferol dimethyl ethers were ascribed to **104**, **109** and **111**, respectively. For example, **111** ([M-H]^−^ at *m*/*z* 313.072, C_17_H_13_O_6_) is used to illustrate the fragmentation pathway of methoxylated derivatives (Figure 2D). (−) ESI-MS/MS **111** yielded typical radical losses at *m*/*z* 298.048 [M-H-•CH_3_]^−^ and 283.025 [M-H-2•CH_3_]^−^, accompanied with successive neutral losses at 255.030 [M-H-2•CH_3_-CO]^−^, 227.034 [M-H-2•CH_3_-2CO]^−^, 211.039 [M-H-2•CH_3_-CO-CO_2_]^−^and 183.044 [M-H-2•CH_3_-2CO-CO_2_]^−^. On the other hand, (+) ESI-MS/MS prominent fragment ions at *m*/*z* 135.044 (^0,2^B^+^) and 121.029 (^0,2^B^+^-CH_2_) pointed out a methoxylated RDA ion ^0,2^B^+^. Regarding **108** ([M-H]^−^ at *m*/*z* 299.056, C_16_H_11_O_6_), an unsubstituted A-ring was deduced from the prominent RDA ions at *m*/*z* 151.002 (^1,3^A^−^), 135.007 (^0,3^A^−^) and 107.012 (^0,4^A^−^), while the ion at *m*/*z* 132.020 (^1,3^B^−^) had a methoxylated B-ring, as was seen in kaempferide (4′methoxy-kaempferol) [30]. In the same manner, peak **110** gave a precursor ion at [M-H]^−^ at *m*/*z* 359.078 (C_18_H_15_O_6_) together with the transitions at *m*/*z* 344.054 → 329.030 →314.008 resulting from the subsequent losses of tree methyl radicals (Appendix A). A methoxylated A-ring was discernable by the prominent RDA ion at *m*/*z* 165.0180 (^1,3^A^−^), instead of 151.003 in myricetin. Additionally, in (+) ESI-MS/MS, a methoxylated B-ring was deduced from the RDA ion at *m*/*z* 167.034 (^0,2^B^+^). Thus, 110 was assigned to the myricetin trimethyl ether. Within this group, the dominance of kaempferol dimethyl ether (2.61%), myricetin *O*-rhamnoside (myricitrin) (1.04%) and naringenin (1.59%) should be noted. The relative abundance of flavonoids accounted for 11.92%. 

### 2.2. Total Phenolic and Flavonoid Content

Phenolic compounds are cornerstones of both the nutraceutical and pharmaceutical sectors. They are important as antioxidants, protecting the body from attacks by free radicals. In addition, they inhibit lipid peroxidation caused by the attack of free radicals in food. In this sense, novel and effective phenolic sources are one of the most attractive topics in the scientific fields. In the present study, we determined the total phenolic and flavonoid content of the methanol-aqueous extract of *P. graveolens* using spectrophotometric assays (Table 2). The total phenolic content was 83.86 mg GAE/g. Flavonoids were the main group of phenols and exhibited important biological activities, including antioxidant, anticancer or antimicrobial properties. The total flavonoid content was determined to be 9.49 mg RE/g. In the literature, we found different values for the total phenolic and flavonoid contents in *P. graveolens* or other *Pelargonium* members. For example, El Ouadi et al. [31] reported that the total phenolic and flavonoid contents of diethyl ether and ethyl acetate extracts were 400–437 µg GAE/mg and 12–29 µg RE/mg, respectively. In another study by Boukrish et al. [32], the total phenolic and flavonoid contents in the different solvents extracts (methanol and water) of leaves and flowers of *P. graveolens* was found to be 54.71–109.76 mg GAE and 21.21–78.49 mg quercetin equivalent (QE). The total phenolic and flavonoid contents were also reported by El Aanachi et al. [33] as 36.55–381.25 mg GAE/g and 18.02–330.08 mg QE/g, respectively; they also reported that methanol extract contained more phenolics compared to n-hexane and dichloromethane. Similarly, Hsouna and Hamdi [34] found that methanol was the most effective solvent for the phenolics and flavonoids of *P. graveolens* compared to n-hexane, ethyl acetate and water. The observed differences can also be explained by geographical and climatic factors. Additionally, there are some concerns with spectrophotometric testing. Over the last decade, several researchers reported that tests did not reflect accurate levels of these compounds in the plant extracts [35,36]. In particular, the Folin–Ciocalteu reagent is not specific to phenols, and therefore, the results obtained may not be entirely accurate. From this point on, spectrophotometric results were confirmed by chromatographic methods, including HPLC or LC-MS/MS.

### 2.3. Antioxidant Properties

Due to the modern lifestyle and diet, people are at an increased risk of attack by free radicals. Free radicals contain one or more unpaired electrons and are therefore very reactive. The reactivity is delivered more effectively and attacks other biomolecules. Thus, the situation can be harmful. Antioxidants can neutralize free radicals and alleviate their harmful effects. They also extend the shelf life of foods by controlling lipid peroxidation. Based on this information, new and effective sources of antioxidants are the most attractive topics in pharmaceutical and nutraceutical research [37]. Since antioxidant mechanisms work in different ways, no uniform method has been described so far. In the current study, we investigated the antioxidant properties of the methanol-aqueous extract of *P. graveolens* using different methods. The results are shown in Table 2. DPPH and ABTS^•+^ are the most commonly used radicals to evaluate antioxidant scavenging ability. In particular, through the transfer of hydrogen from antioxidants, the radicals can be intercepted, and the changes can be determined using spectrophotometric measurements. The DPPH and ABTS scavenging abilities were reported to be 273.45 mg TE/g and 531.97 mg TE/g, respectively. The radical scavenging ability of *P. graveolens* has been reported in the literature. For example, Zengin et al. [38] reported that the best radical scavenging abilities in DPPH and ABTS^•+^ assays were found in ethanol (DPPH: 713.03 mg TE/g; ABTS^•+^: 1170.78 mg TE/g) and water extracts (DPPH: 774.35 mg TE/g; ABTS^•+^: 1257.28 mg TE/g) of *P. endlicherianum*. In another study by Ennaifer et al. [39], the DPPH radical scavenging abilities of decoction (113.86 mg TE/g) and infusion (47.31 mg TE/g) of *P. graveolens* were lower than those reported in the current study. Checkouri et al. [40] also reported significant DPPH radical ability for infusion and decoction of *P. graveolens* (inhibition ratio > 50%). Reducing the power provides information about the electron donation ability of antioxidants. A strong ability to donate electrons is accompanied by a strong antioxidant effect [41]. For this purpose, CUPRAC and FRAP assays were performed, and the results were 431.32 mg TE/g and 292.21 mg TE/g, respectively. El Aanachi et al. [33] investigated the free radicals-reducing abilities of different extracts of *P. graveolens* using CUPRAC (EC_50_: 20.29 µg/mL) and FRAP (EC_50_: 43.38 µg/mL), and the best action was noted for methanol extract in both assays. The phosphomolybdenum assay also involves the reduction of Mo (VI) to Mo (V) by antioxidants under acidic conditions. Recently, this assay has gained interest due to its simplicity and cost-effectiveness [41]. As shown in Table 2, the result was 2.71 mmol TE/g of extract. Metal chelation is associated with controlling the production of hydroxyl radicals, the most dangerous radical. In the current study, a ferrozine assay was performed, and the ability was found to be 13.44 mg EDTAE/g. Based on the structure activity findings, *Pelargonium* species are rich in antioxidant compounds, including phenolic acids and tannins, and the observed antioxidant abilities may be attributed to the presence of these compounds [42,43,44,45,46]. Indeed, gallic, protocatechuic and caffeic acid; kaempferol; myricetin; and isorhamnetin provide effective protection against oxidative damage by reactive species [47,48,49]. It is worth noting that the caffeic acid conjugated possessed higher antiradical activity towards superoxide and hydroxyl radicals than caffeic acid [50]. Importantly, caftaric acid induces antioxidant pathways, leading to increased levels of the antioxidant enzymes superoxide dismutase (SOD), catalase (CAT) and glutathione reductase, which provided antioxidant defense in a pathological model [51]. Caftaric acid has a marked impact on the elevation of the oxidant enzyme inducible nitric oxide synthase (iNOS) and cytokine expression (IL-6, TNF-α) in inflammation injuries. Numerous studies have demonstrated that caftaric-acid-rich lettuce and grape cultivars, grape juices and wines mitigate lipid peroxydation and reactive oxygen species, which may hold significance for reducing increased oxidative stress [50,52]. Various galloyl-glucopyranoses (mono-, di-, tri-, tetra- and penta-*O*-galloyl-glucopyranose) possess antioxidant properties [53]. A higher number of galloyl groups have an increased ability to scavenge free radicals. Flavanols prevent oxidative stress by direct scavenging of free radicals, metal chelation, reduction of tocopheryl radicals, and induction of antioxidant enzymes as well as phase II detoxifying enzymes such as glutathione-S-transferase [13].

### 2.4. Enzyme-Inhibitory Properties

Enzymes are versatile players in both biochemical reactions and pharmaceutical applications. In the last decade, most drugs (>50%) contained at least one enzyme inhibitor and could help treat some global diseases such as Alzheimer’s disease, diabetes and cardiovascular problems [54]. This situation is due to the inhibition of key enzymes, alleviating pathological observations. For example, inhibiting acetylcholinesterase can increase acetylcholine levels in the synaptic cleft and help to improve memory function in Alzheimer’s patients [55]. In addition, inhibiting pancreatic lipase can prevent the hydrolysis of lipids in the gastrointestinal system, thereby controlling obesity [56]. For this purpose, various compounds are synthesized as enzyme inhibitors in the pharmaceutical industry. But when taken for a long time, most of them show unpleasant side effects. Therefore, we are looking for alternative and effective inhibitors to replace them.

Based on the above facts, we tested the enzyme-inhibitory effect of methanol-aqueous extract of *P. graveolens* against cholinesterase, tyrosinase, amylase, glucosidase and lipase. The extract showed significant cholinesterase-inhibitory activity, and the results were 2.80 mg GALAE/g and 2.20 mg GALAE/g for AChE and BChE, respectively. In another study by Zengin et al. [38], the ethanol extract of *P. endlicherianum* exhibited the best AChE (3.74 mg GALAE/g)- and BChE (1.10 mg GALAE/g)-inhibitory effect compared to other extracts (enzyme concentrations were identical to the present study). The AChE-inhibitory effect reported was higher than ours; however, our BChE value surpassed the reported one. Tyrosinase is a key enzyme in melanin synthesis and is used to control hyperpigmentation problems. The tested extract had a tyrosinase-inhibiting effect, with 75.49 mg CAE/g. In a previous study conducted by El Aanachi [33], the methanol extract of *P. graveolens* exhibited the strongest anti-tyrosinase effect, with the lowest IC_50_ value (21.11 µg/mL) (enzyme concentration was 150 units/mL). In addition, the best tyrosinase-inhibitory value was also reported for ethanol extract of *P. endlicherianum* (64.66 mg KAE/g) in another study [38] (enzyme concentrations were identical to the present study). Amylase and glucosidase are known as antidiabetic enzymes, and the extract had an inhibitory effect on both (amylase: 0.79 mmol ACAE/g; glucosidase: 3.75 mmol ACAE/g). The value for amylase inhibition was very close to that of the ethanol extract of *P. endlicherianum* (0.80 mmol ACAE), while our glucosidase-inhibition value was greater than [38] (enzyme concentrations were identical to the present study). The extract also had an inhibitory effect on pancreatic lipase at 28.91 mg OE/g. The presence of some compounds in the extract may also have contributed to the observed enzyme-inhibitory effect. For instance, gallic acid, ellagic acid and rutin are considered important inhibitors against the enzymes tested [57,58,59,60,61,62,63,64,65,66]. Caffeic acid derivates such as caftaric acid exhibited increased inhibition ability towards α-glucosidase [67]. Significant advances have been made in understanding the anti-diabetic effects underlying caftaric-acid-mediated activation of insulin secretion [68] by modulating the gene expression profile of key insulin regulatory genes and glucose transporter 2 (GLUT2). Inhibition of GLUT2 is an essential target for glycemia control to avoid an increase in post-prandial glucose. Flavan-3-ols, flavanols (quercetin, isorhamnetin, myricetin), naringenin and phenolic acids (caffeic and ferulic acids) can limit glucose absorption via interaction with GLUT-2 [67,69]. Caffeoylisocitric acid elicits activation of the nuclear factor erythroid 2-related factor 2 (Nrf2) antioxidant pathway, which mitigates oxidative stress resulting from high glucose levels in mesangial cells [70]. Additionally, recent review articles have emphasized the beneficial effects of protocatechuic acid and vanillic acid in a number of experimental models for the prevention of diabetes and neurodegenerative diseases, including Alzheimer’s [71,72]. Besides, *p*-coumaric acid is associated with elevation of antioxidant enzymes and the prevention and improvement of diabetes and neuroprotection [73].

## 3. Materials and Methods

### 3.1. Plant Material

*P. graveolens* seedlings were provided by the greenhouse “Zelena prolet” (Sofia, Bulgaria). Plant cultivation was carried out in an herbal garden in plastic pots with the following fertilizer mixture: 50% alluvial peat, 30% soil mixture, 5% mineral perlite, 5% coconut fiber, 10% zeolite, 0.08% N, 0.06% P, 0.09% K and pH 6.5 (GAMMA LTD, Sofia, Bulgaria). The species’ taxonomic identity was confirmed by R. Gevrenova according to http://www.worldfloraonline.org/ (accessed on: 1 January 2024). The voucher specimen was deposited at Herbarium Academiae Scientiarum Bulgariae (SOM 179216). The harvesting of the leaves was performed during the ramification of well-developed stalks in June 2023. Throughout the gardening period, no herbicides were used to preserve the content of the chemical compounds identified. The plant material (leaves) was dried in the laboratory for one week at room temperature (20–22 °C) and 50% relative humidity. A low-temperature drying procedure (20–22 °C) was used so that the water was not completely removed from the plant material, which would interfere with further analysis due to changes in the surface activity of the stationary phase [74]. Moreover, the leaves were dried until a constant weight was obtained for the plant material [75]. As the fresh/dried mass ratio of *P. graveolens* leaves is 1:6 [76], 600 g of fresh biomass of rose geranium leaves was used to obtain 100 g of dry plant material. Then, it was comminuted with a grinder (Rohnson, R-942, 220–240 V, 50/60 Hz, 200 W, Prague, Czech Republic), and the powdered plant material was stored in a cool, dry place until further examination.

### 3.2. Sample Extraction 

Air-dried powdered leaves (100 g) were extracted with 80% MeOH (1:20 *w*/*v*) by sonication (100 kHz, ultra-sound bath Biobase UC-20C, Biobase, Jinan, Shandong, China) for 15 min (×2) at room temperature. The methanol was evaporated in vacuo (35 °C), and water residues were lyophilized (lyophilizer Biobase BK-FD10P, Biobase, Jinan, Shandong, China; −65 °C) to yield a crude extract of 15.6 g. Then, the lyophilized extracts were dissolved in 80% methanol (0.1 mg/mL), filtered through a 0.45 μm syringe filter (Polypure II, Alltech, Lokeren, Belgium), and an aliquot (2 mL) of each solution was subjected to UHPLC–HRMS analyses [74]. The same extracts were used for biological tests.

### 3.3. Chemicals

Acetonitrile (hypergrade for LC–MS), formic acid (for LC–MS) and methanol (analytical grade) were purchased from Chromasolv (Sofia, Bulgaria). The reference standards used for compound identification were obtained from Extrasynthese (Genay, France) (for gallic, protocatechuic, 4-hydroxybenzoic, 3- hydroxybenzoic, o-, m-, and p-coumaric, ferulic, caffeic, gentisic, vanillic, and salicylic acids, vanillin, rutin, myricetin, myricitrin, hyperoside, isoquercitrin, kaempferol 3-*O*-rutinoside, isorhamnetin 3-*O*-rutinoside, kaempferol 3-*O*-glucoside, isorhamnetin 3-*O*-glucoside, quercetin, isorhamnetin, kaempferol) and Phytolab (Vestenbergsgreuth, Bavaria, Germany) (caftaric and ellagic acids, eriodyctiol and naringenin). A working solution containing 0.1 mg/mL of the assayed compounds was prepared from a stock solution in methanol containing 0.5 mg/mL. 

The chemicals for antioxidant and enzyme inhibition assays were purchased from Sigma-Aldrich (Darmstadt, Germany). They were ABTS, DPPH, gallic acid, rutin, electric eel acetylcholinesterase (AChE) (type-VI-S, EC 3.1.1.7), horse serum butyrylcholinesterase (BChE) (EC 3.1.1.8), galantamine, acetylthiocholine iodide (ATChI), butyrylthiocholine chloride (BTChI) 5,5-dithio-bis(2-nitrobenzoic) acid (DTNB), tyrosinase (EC1.14.18.1, mushroom), glucosidase (EC. 3.2.1.20, from *Saccharomyces cerevisiae*), amylase (EC. 3.2.1.1, from porcine pancreas), sodium molybdate, sodium carbonate, Folin–Ciocalteu reagent, hydrochloric acid, sodium hydroxide, trolox, ethylenediaminetetraacetate (EDTA), neocuproine, cupric chloride, ammonium acetate, ferric chloride, 2,4,6-Tris(2-pyridyl)-s-triazine (TPTZ), ammonium molybdate, ferrozine, ferrous sulphate hexahydrate, kojic acid and acarbose. All chemicals were of analytical grade.

### 3.4. UHPLC-HRMS 

The UHPLC-HRMS analyses were performed as previously described [27] on a Q Exac-tive Plus mass spectrometer (ThermoFisher Scientific, Inc., Waltham MA, USA) equipped with a heated electrospray ionization (HESI-II) probe (ThermoScientific). The equipment was operated in negative and positive ion modes within the *m*/*z* range from 150 to 1500. The chromatographic separation was achieved on a reversed phase column Kromasil EternityXT C18, Nouryon, Amsterdam, The Netherlands (1.8 µm, 2.1 × 100 mm) at 40 °C. The UHPLC analyses were run with a mobile phase containing 0.1% formic acid in water (A) and 0.1% formic acid in acetonitrile (B). The run time was 33 min. The flow rate was 0.3 mL/min. The gradient elution program was used as follows: 0–1 min, 0–5% B; 1–20 min, 5–30% B; 20–25 min, 30–50% B; 25–30 min, 50–70% B; 30–33 min, 70–95%; 33–34 min 95–5% B. Equilibration time was 4 min. The injection volume and the flow rate were set to 1 µL and 300 µL/min, respectively. Data were processed by the Xcalibur 4.2 (ThermoScientific, Waltham, MA, USA) instrument control/data handling software and MZmine 2 software.

### 3.5. Assay for Total Phenolic and Flavonoid Contents

According to the methods specified by [77], total phenolics and flavonoids were quantified. The extract was prepared at a concentration of 2 mg/mL. Gallic acid (GA) and rutin (RE) served as standards in the assays, and the outcomes were reported as gallic acid equivalents (GAE) and rutin equivalents. All experimental details are given in the Appendix A. 

### 3.6. Assays for In Vitro Antioxidant Capacity

For the DPPH (1,1-diphenyl-2-picrylhydrazyl) radical scavenging assay, the sample solution was added to 4 mL of a 0.004% methanol solution of DPPH. The sample absorbance was 517 nm after 30 min incubation at room temperature in the dark. Trolox was used as a standard (0.01–01 mg/mL) and DPPH radical scavenging activity was expressed as milligrams of trolox equivalents (mg TE/g extract) [78].

For the ABTS^•+^ (2,2′-azino-bis(3-ethylbenzothiazoline) 6-sulfonic acid) radical scavenging assay, briefly, ABTS^•+^ was produced directly by reacting 7 mM ABTS^•+^ solution with 2.45 mM potassium persulfate and allowing the mixture to stand for 12–16 min in the dark at room temperature. Prior to beginning the assay, the ABTS^•+^ solution was diluted with methanol to an absorbance of 0.700 ± 0.02 at 734 nm. The sample solution was added to the ABTS^•+^ solution (2 mL) and mixed. The sample absorbance was read at 734 nm after a 30 min incubation at room temperature. Trolox was used as a standard (0.01–01 mg/mL), and the ABTS^•+^ radical scavenging activity was expressed as milligrams of trolox equivalents (mg TE/g extract) [79].

For the CUPRAC (cupric ion reducing activity) activity assay, the sample solution was added to a premixed reaction mixture containing CuCl_2_ (1 mL, 10 mM), neocuproine (1 mL, 7.5 mM) and an NH_4_Ac buffer (1 mL, 1 M, pH 7.0). Similarly, a blank was prepared by adding the sample solution (0.5 mL) to the premixed reaction mixture (3 mL) without CuCl_2_. Then, the sample and blank absorbances were read at 450 nm after a 30 min incubation at room temperature. The absorbance of the blank was subtracted from that of the sample. Trolox was used as a standard (0.01–01 mg/mL), and CUPRAC activity was expressed as milligrams of trolox equivalents (mg TE/g extract) [80].

For the FRAP (ferric reducing antioxidant power) activity assay, the sample solution was added to the premixed FRAP reagent (2 mL) containing an acetate buffer (0.3 M, pH 3.6), 2,4,6-tris(2-pyridyl)-S-triazine (TPTZ) (10 mM) of 40 mM HCl and ferric chloride (20 mM) in a ratio of 10:1:1 (*v/v/v*). Then, the sample absorbance was read at 593 nm after a 30 min incubation at room temperature. Trolox was used as a standard (0.01–01 mg/mL), and FRAP activity was expressed as milligrams of trolox equivalents (mg TE/g extract) [81].

For the phosphomolybdenum method, the sample solution was combined with 3 mL of the reagent solution (0.6 M sulfuric acid, 28 mM sodium phosphate and 4 mM ammonium molybdate). The sample absorbance was read at 695 nm after a 90 min incubation at 95 °C. Trolox was used as a standard (0.1–1 mg/mL), and the total antioxidant capacity was expressed as millimoles of trolox equivalents (mmol TE/g extract) [82].

For the metal chelating activity assay, briefly, the sample solution was added to the FeCl_2_ solution (0.05 mL, 2 mM). The reaction was initiated by the addition of 5 mM ferrozine (0.2 mL). Similarly, a blank was prepared by adding the sample solution (2 mL) to the FeCl_2_ solution (0.05 mL, 2 mM) and water (0.2 mL) without ferrozine. Then, the sample and blank absorbances were read at 562 nm after 10 min incubation at room temperature. The absorbance of the blank was subtracted from that of the sample. EDTA was used as a chelator (10–40 µg/mL), and the metal chelating activity was expressed as milligrams of EDTA (disodium edetate) equivalents (mg EDTAE/g extract) [83].

### 3.7. Inhibitory Effects against Some Key Enzymes

Enzyme inhibition experiments on the samples were conducted following established protocols [84]. The extract was prepared at a concentration of 0.1–2 mg/mL. Amylase and glucosidase inhibition were quantified in acarbose equivalents (ACAE) per gram of extract, while acetylcholinesterase (AChE) and butyrylcholinesterase (BChE) inhibition were expressed in milligrams of galanthamine equivalents (GALAE) per gram of extract. Tyrosinase inhibition was measured in milligrams of kojic acid equivalents (KAE) per gram of extract. Lipase inhibition was evaluated as the equivalent of orlistat (OE) per gram of extract. All experimental details are given in the Appendix A.

### 3.8. Statistical Analysis

The experiments were performed in triplicate, and the results were presented as the mean and standard deviation. GraphPad 9.1 was used to evaluate the obtained results. 

## 4. Conclusions

This study allowed for obtaining an in-depth phytochemical profiling of rose geranium methanol-aqueous extract for the first time. More than 110 secondary metabolites, notably, hydroxybenzoic and hydroxycinnamic acids and their glycosides, acyltartaric, acylcitric/acylisocitric acids, gallotannins along with flavanols, flavanones and flavan-3-ols were dereplicated/annotated. The profile was dominated by caffeoylcitric/caffeoylisocitric acid, caftaric acid and its glycoside methylgalate, gallic acid-hexosides, tartaric, malic and citric/isocitric acids. Myricetin *O*-rutinoside, myricitrin and kaempferol methyl and dimethyl ether appeared to be characteristic for rose geranium leaves. The strong antioxidant potential (DPPH, ABTS^•+^, FRAP and CUPRAC) of the leaves could be related to the presence of myricetin- and quercetin glycosides together with the gallic acid derivatives. Indeed, methoxylated flavanols and related derivatives could be associated with anti-BChE, anti-AChE and anti-tyrosinase activity. Caftaric acid along with hydroxycinnamic acids conjugates and phenolic acid-hexosides accounted for the α-glucosidase-inhibitory activity. In addition to provoking an antioxidant response, rose geranium extract exhibited in vitro inhibitory activity towards key enzymes of the neurotransmission and carbohydrate metabolism, which generates further interest in the herbal drug as potential therapeutic candidate for corresponding diseases. The current study argues for work geared towards additional in vivo experiments of the beneficial effects of a *P. graveolens* extract on lipid and glucose metabolism and antioxidant status in animal models with induced metabolic dysfunction-associated fatty liver disease. 

## Figures and Tables

**Figure 1 plants-13-02612-f001:**
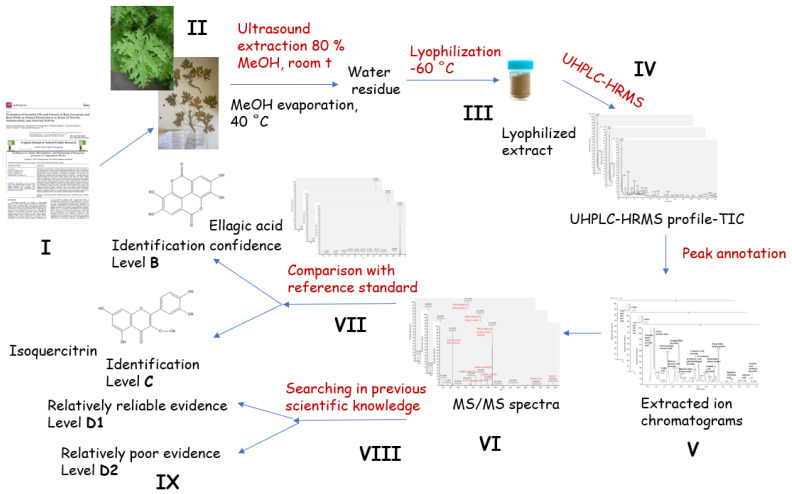
The workflow for the essential aspects in qualitative plant metabolite profiling, including critical review of previous literature (**I**); cultivated material documented by vouchers in public herbaria (**II**); gentle extraction procedure avoiding artefacts (**III**); state-of-the-art chemical analysis (**IV**); careful chemical data interpretation to obtain extracted ion chromatograms (**V**); analyzing MS/MS fragmentation pathway of each compound (**VI**); parallel comparison of tR, exact high-resolution mass and MS/MS spectra with reference standards (**VII**) or comparison with previous scientific knowledge (**VIII**); and metabolite identification assigning reliable confidence levels (**IX**).

**Table 1 plants-13-02612-t001:** Secondary metabolites in *Pelargonium graveolens* methanol-aqueous extracts.

№	Identified/Tentatively Annotated Compound	Molecular Formula	Exact Mass [M-H]^−^	t_R_ (min)	Identification Confidence Level (Çiçek et al. [18])
**Hydroxybenzoic, Hydroxycinnamic Acids and Their Glycosides, Coumarins and Aliphatic Acids**					
1.	tartaric acid	C_4_H_6_O_6_	149.0092	0.70	D1
2.	mallic acid	C_4_H_6_O_5_	133.0143	0.74	D1
3.	citric/isocitric acid	C_6_H_8_O_7_	191.0197	0.75	D1
4.	citric/isocitric acid	C_6_H_8_O_7_	191.0197	0.90	D1
5.	gallic acid *	C_7_H_6_O_5_	169.0143	1.15	B
6.	protocatechuic acid-*O*-dihexoside	C_19_H_26_O_14_	477.1250	1.52	D1
7.	protocatechuic acid-*O*-hexoside 1	C_13_H_16_O_9_	315.0727	1.67	D1
8.	vanillic acid-*O*-hexoside 1	C_14_H_18_O_9_	329.0875	1.77	D1
9.	protocatechuic acid *	C_7_H_6_O_4_	153.0181	2.03	B
10.	protocatechuic acid-*O*-hexoside 2	C_13_H_16_O_9_	315.0727	2.10	D1
11.	hydroxybenzoic acid-*O*-hexoside 1	C_13_H_16_O_8_	299.0778	2.15	D1
12.	syringic acid-*O*-hexoside	C_15_H_20_O_10_	359.0985	2.26	D1
13.	vanilloyl-*O*-hexose	C_14_H_18_O_9_	329.0875	2.47	D1
14.	dihydrocaffeic acid-*O*-hexoside	C_15_H_20_O_9_	343.1035	2.49	D2
15.	coumaric acid-*O*-hexoside 1	C_15_H_18_O_8_	325.0930	2.51	D1
16.	caffeic acid–*O*-hexoside 1	C_15_H_18_O_9_	341.0871	2.63	D1
17.	*p*-hydroxyphenylacetic acid *O*-hexoside 1	C_14_H_18_O_8_	313.0929	2.66	D1
18.	aesculetin-*O*-hexoside	C_15_H_15_O_9_	339.0724	2.69	D1
19.	syringyl-*O*-hexose	C_15_H_20_O_10_	359.0984	2.76	D1
20.	4-hydroxybenzoic acid *	C_7_H_6_O_3_	137.0230	2.84	D1
21.	3-hydroxybenzoic acid *	C_7_H_6_O_3_	137.0230	2.99	D1
22.	hydroxybenzoic acid-*O*-hexoside 2	C_13_H_16_O_8_	299.0778	2.99	D1
23.	caffeic acid O-hexoside 2	C_15_H_18_O_9_	341.0871	3.08	D1
24.	*p*-coumaric acid *	C_9_H_8_O_3_	163.0389	3.10	B
25.	scopoletin *O*-hexoside	C_16_H_18_O_9_	353.0878	3.16	D1
26.	dihydrocaffeic acid-O-hexoside	C_15_H_20_O_9_	343.1035	3.19	D1
27.	*p*-hydroxyphenylacetic acid *O*-hexoside 2	C_14_H_18_O_8_	313.0929	3.30	D1
28.	coumaric acid-*O*-hexoside 2	C_15_H_18_O_8_	325.0930	3.34	D1
29.	*m*-coumaric acid *	C_9_H_8_O_3_	163.0389	3.35	B
30.	vanillic acid-*O*-hexoside 2	C_14_H_18_O_9_	329.0875	3.39	D1
31.	vanillyl alcohol-(acetyl)-hexoside	C_16_H_22_O_9_	357.1191	3.41	D1
32.	aesculetin	C_9_H_6_O_4_	177.0193	3.45	D1
33.	scopoletin O-hexoside isomer	C_16_H_18_O_9_	353.0878	3.48	D1
34.	ferulic acid *	C_10_H_10_O_4_	193.0494	3.55	B
35.	caffeic acid *	C_9_H_8_O_4_	179.0339	3.56	B
36.	gentisic acid *	C_7_H_6_O_4_	153.0180	3.67	B
37.	vanillic acid-*O*-hexoside 3	C_14_H_18_O_9_	329.0875	4.22	D1
38.	vanillin *	C_8_H_8_O_3_	151.0401	4.34	B
39.	phenylethyl-*O*-pentosylhexoside (primeveroside)	C_20_H_30_O_12_	461.1665	4.48	D1
40.	*o*-coumaric acid *	C_9_H_8_O_3_	163.0389	4.55	B
41.	coumaric acid-*O*-hexoside 3	C_15_H_18_O_8_	325.0929	4.70	D1
42.	phenylethyl-*O*-pentosylhexoside (primeveroside)	C_20_H_30_O_12_	461.1665	4.75	D1
43.	vanillic acid *	C_8_H_8_O_4_	167.0338	4.79	B
44.	scopoletin–(caffeoyl)-hexoside	C_25_H_26_O_13_	533.1301	5.12	D1
45.	glansreginic acid *O*-hexoside	C_18_H_28_O_10_	403.1609	5.43	D2
46.	digalloylcitramalic acid	C_19_H_18_O_14_	469.0624	5.52	D2
47.	salicylic acid *	C_7_H_6_O_3_	137.0230	6.28	B
48.	ferulic acid-(vanillyl)-hexoside	C_24_H_26_O_12_	505.1352	6.45	D1
49.	gentisic acid-(feruloyl)-hexoside	C_23_H_24_O_12_	491.1195	6.71	D1
**Acyltartaric Acids**					
50.	caftaric acid *	C_13_H_12_O_9_	311.0409	2.16	C
51.	caffeoyltartaric acid-hexoside-	C_19_H_22_O_14_	473.0937	2.28	D2
52.	caftaric acid isomer	C_13_H_12_O_9_	311.0409	2.36	D1
53.	cafeoyltartaric acid dimer	C_26_H_23_O_18_	623.0892 [2M-H]^−^	2.36	D2
54.	dicaffeoyltartaric acid 1	C_22_H_20_O_13_	491.0831	2.58	D2
55.	dicaffeoyltartaric acid 2	C_22_H_20_O_13_	491.0831	2.79	D2
56.	coumaroyltartaric acid	C_13_H_12_O_8_	295.0459	3.10	D2
57.	feruloyltartaric acid	C_14_H_14_O_9_	325.0565	3.55	D2
**Citric/Isocitric Acid Esters (Acylcitric Acids)**					
58.	caffeoylcitric/isocitric acid 1	C_15_H_14_O_10_	353.0514	2.41	D1
59.	caffeoylcitric/isocitric acid 2	C_15_H_14_O_10_	353.0514	2.76	D1
60.	caffeoylcitric/isocitric acid 3	C_15_H_14_O_10_	353.0514	2.90	D1
61.	caffeoylcitric/isocitric acid 4	C_15_H_14_O_10_	353.0514	3.28	D1
62.	caffeoylcitric/isocitric acid 5	C_15_H_14_O_10_	353.0514	3.95	D1
63.	coumaroylcitric/isocitric acid 1	C_15_H_14_O_9_	337.0565	4.16	D1
64.	caffeoylcitric/isocitric acid 6	C_15_H_14_O_10_	353.0514	4.29	D1
65.	coumaroylcitric/isocitric acid 2	C_15_H_14_O_9_	337.0565	4.85	D2
66.	coumaroylcitric/isocitric acid 3	C_15_H_14_O_9_	337.0565	5.26	D2
67.	feruloylcitric/isocitric acid 1	C_16_H_16_O_10_	367.0671	5.30	D2
68.	feruloylcitric/isocitric acid 2	C_16_H_16_O_10_	367.0671	5.63	D2
**Gallotannins**					
69.	galloyl-*O*-hexoside 1	C_13_H_16_O_10_	331.0671	1.22	D1
70.	galloyl-*O*-hexoside 2	C_13_H_16_O_10_	331.0671	1.58	D1
71.	gallocatechin	C_15_H_14_O_7_	305.0667	1.80	D1
72.	digalloyl-O-hexose	C_20_H_20_O_14_	483.0780	2.99	D1
73.	digallic acid	C_14_H_10_O_9_	321.0252	3.10	D1
74.	methylgallate	C_8_H_8_O_5_	183.0299	3.16	D1
75.	tetragalloyl-hexoside	C_34_H_28_O_22_	787.1000	4.94	D1
76.	ellagic acid *	C_14_H_6_O_8_	300.9990	5.01	B
**Flavonoids**					
77.	eriodyctiol *O*-hexoside	C_21_H_22_O_11_	449.1089	4.03	D1
78.	myricitin 3-*O*-pentosylhexoside	C_26_H_28_O_17_	611.1254	4.18	D1
79.	myricetin 3-*O*-rutinoside	C_27_H_30_O_17_	625.1410	4.48	D1
80.	quercetin 3-*O*-pentosylhexoside	C_26_H_28_O_16_	595.1305	4.71	D1
81.	myricetin *O*-pentoside	C_20_H_18_O_12_	449.0726	4.96	D1
82.	rutin *	C_27_H_30_O_16_	609.1464	5.08	C
83.	myricetin *O*-rhamnoside (myricitrin)	C_21_H_20_O_12_	463.0885	5.11	C
84.	hyperoside *	C_21_H_20_O_12_	463.0885	5.19	C
85.	quercetin *O*-hexuronide	C_22_H_22_O_12_	477.1039	5.23	D1
86.	kaempferol *O*-pentosylhexoside	C_26_H_28_O_15_	579.1355	5.27	D1
87.	Isoquercitrin *	C_21_H_20_O_12_	463.0885	5.30	C
88.	myricetin methylether *O*-hexoside	C_22_H_22_O_13_	493.0988	5.35	D1
89.	phloretin 3′, 5–diC-hexoside	C_27_H_34_O_15_	597.18249	5.37	D1
90.	kaempferol *O*-deoxyhexosyl-*O*-hexoside	C_27_H_30_O_15_	593.1512	5.41	D1
91.	quercetin 3-*O*-pentoside	C_20_H_18_O_11_	433.0776	5.64	D1
92.	kaempferol 3-*O*-rutinoside *	C_27_H_30_O_15_	593.1512	5.64	C
93.	phloretin C-hexoside	C_21_H_24_O_10_	435.1297	5.74	D1
94.	isorhamnetin 3-*O*-rutinoside *	C_28_H_32_O_17_	623.1618	5.78	C
95.	kaempferol 3-*O*-glucoside *	C_21_H_19_O_11_	447.0934	5.88	C
96.	isorhamnetin 3-*O*-glucoside *	C_22_H_22_O_12_	477.1044	6.03	C
97.	naringenin 7-*O*-hexoside	C_21_H_20_O_10_	431.0984	6.06	D1
98.	kaempferol 3-O-pentoside	C_20_H_18_O_10_	417.0827	6.07	D1
99.	eriodyctiol *	C_15_H_12_O_6_	287.0561	6.29	B
100.	myricetin *	C_15_H_10_O_8_	317.0303	6.29	B
101.	luteolin 7-*O*-caffeoylhexoside	C_30_H_26_O_14_	609.1250	7.01	D1
102.	quercetin *	C_15_H_10_O_7_	301.0354	7.61	B
103.	isorhamnetin *	C_16_H_12_O_7_	315.0512	8.09	B
104.	myricetin dimethyl ether	C_17_H_14_O_8_	345.0616	8.11	D1
105.	naringenin *	C_15_H_12_O_5_	271.0612	8.58	B
106.	kaempferol dimethyl ether *O*-hexoside	C_31_H_22_O_8_	521.1242	8.79	D1
107.	kaempferol *	C_15_H_10_O_6_	285.0405	8.84	B
108.	kaempferol methyl ether (kaempferide)	C_16_H_12_O_6_	299.0561	9.33	D1
109.	quercetin dimethyl ether	C_17_H_14_O_7_	329.0677	9.61	D1
110.	myricetin trimethyl ether	C_18_H_16_O_8_	359.0772	10.68	D1
111.	kaempferol dimethyl ether	C_17_H_14_O_6_	313.0718	12.30	D1
**Other Compounds**					
112.	bergenin *O*-coumaric acid	C_23_H_22_O_11_	473.1089	8.03	D1

* Identified by comparison with standard reference.

**Table 2 plants-13-02612-t002:** Total bioactive compounds, antioxidants and enzyme-inhibitory properties of the tested extract.

Parameters	Results
*Total bioactive compounds*	
Total phenolic content (mg GAE/g)	83.86 ± 1.67
Total flavonoid content (mg RE/g)	9.49 ± 0.26
*Antioxidant properties*	
DPPH scavenging ability (mg TE/g)	273.45 ± 4.31
ABTS^•+^ scavenging ability (mg TE/g)	531.97 ± 10.97
CUPRAC (mg TE/g)	413.22 ± 7.22
FRAP (mg TE/g)	292.21 ± 4.77
Metal chelating (mg EDTAE/g)	13.44 ± 0.44
Phosphomolybdenum (mmol TE/g)	2.71 ± 0.15
*Enzyme-inhibitory properties*	
AChE inhibition (mg GALAE/g)	2.80 ± 0.02
BChE inhibition (mg GALAE/g)	2.20 ± 0.11
Tyrosinase inhibition (mg KAE/g)	75.49 ± 0.49
Amylase inhibition (mmol ACAE/g)	0.79 ± 0.01
Glucosidase inhibition (mmol ACAE/g)	3.75 ± 0.01
Lipase inhibition (mg OE/g)	28.91 ± 4.84

Values are reported as mean ± SD of three parallel measurements. GAE: Gallic acid equivalent; RE: Rutin equivalent; TE: Trolox equivalent; EDTAE: EDTA equivalent; GALAE: Galanthamine equivalent; KAE: Kojic acid equivalent; ACAE: Acarbose equivalent; OE: Orlistat equivalent.

## Data Availability

The original contributions presented in the study are included in the article, further inquiries can be directed to the corresponding author.

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
