# Peer review of "Pelargonium graveolens: Towards In-Depth Metabolite Profiling, Antioxidant and Enzyme-Inhibitory Potential"

_plants, 2024, doi:10.3390/plants13182612_

Round 1

Reviewer 1 Report

Comments and Suggestions for Authors

The manuscript title “Pelargonium graveolens: towards in-depth metabolite profiling, antioxidant and enzyme inhibitory potential” is conducted well and has scientific worth. I have several suggestions to improve the current version of manuscript, it needs significant improvements.

Summary of Manuscript:

Pelargonium graveolens L’Hèr. is valued for its traditional uses as a flavoring, ornamental, and medicinal plant, prompting an in-depth study of its phytochemical profile and antioxidant properties. UHPLC-HRMS analysis identified over 110 secondary metabolites, including acyltartaric and acylcitric acids, gallotannins, flavonoids, and phenolic acids, with acylcitric acids being reported in this species for the first time. The methanol-aqueous extract demonstrated significant antioxidant activity, scavenging DPPH and ABTS radicals, and exhibiting high reducing power in CUPRAC and FRAP assays. The extract showed metal chelating capacity and high performance in the phosphomolybdenum assay, indicating strong antioxidant potential. It also exhibited notable enzyme inhibition, affecting acetyl- and butyrylcholinesterase, tyrosinase, α-glucosidase, α-amylase, and lipase, suggesting potential applications in pharmaceutical and nutraceutical products.

Comments for authors are as follows:

1-      The figure 2 contents are too small unable/hard to read; I suggest authors to provide the figure 2 with large front size.

2-      Table 2: authors written “Total flavonoid content content (mg RE/g)” delete the repeated words.

3-      Line 463-467: how many replications were used? How many number of leaves were t=collected for metabolic determination?

4-      Line 464-465: “The plant material (leaves) was dried in the laboratory for one week at room temperature (20-22°C) and 50% of relative humidity.” Why authors dry the leave at room temperature for one week??? Does this drying procedure may change the inter secondary metabolic profiles. Please provide a strong reference for using this method to dry the leaves at room temperature.

5-      Line 470: “Air-dried powdered leaves (100 g) were extracted” Is it 100 g??? Why to high> maybe it is 100 mg; how many replications?

6-       Line 470-476: where is reference?

7-      Line 479: “The reference standards” write the standard name and concentration and amount of standard used…..

8-      “3.5. Assay for total phenolic and flavonoid contents According to the methods specified by [73], total phenolics and flavonoids were quantified. Gallic acid (GA) and rutin (RE) served as standards in the assays, and the out- comes were reported as gallic acid equivalents (GAE) and rutin equivalents.” Write the full details of phenolic and flavonoid assay. How samples were extracted??? how many sample was weighted ?? What was the standard concentration?? What was the control?? Chemical used in these assay were purchased from which company??? Writing just reference is not enough….

9-      Same questions “3.6. Assays for in vitro antioxidant capacity

10-   According to the methods provided by [74], antioxidant tests were executed. The DPPH, 506 ABTS radical scavenging, CUPRAC, and FRAP test results were expressed as milligrams of Trolox equivalents (TE) per gram of extract. The antioxidant potential determined by the phosphomolybdenum (PBD) assay was measured in millimoles of Trolox equivalents (TE) per gram of extract, and metal chelating activity (MCA) was conveyed as milligrams of disodium edetate equivalents (EDTAE) per gram of extract.” Write the full details of phenolic and flavonoid assay. How samples were extracted??? how many sample was weighted ?? What was the standard concentration?? What was the control?? Chemical used in these assay were purchased from which company??? Writing just reference is not enough….

11-   Same questions “”Write the full details of phenolic and flavonoid assay. How samples were extracted??? how many sample was weighted ?? What was the standard concentration?? What was the control?? Chemical used in these assay were purchased from which company??? Writing just reference is not enough….

12-   Which statistical program was used to analysis the data; write the details in the methods section.

13-   Move the table 1 to supplementary files.

Author Response

Reviewer 1

The manuscript title “Pelargonium graveolens: towards in-depth metabolite profiling, antioxidant and enzyme inhibitory potential” is conducted well and has scientific worth. I have several suggestions to improve the current version of manuscript, it needs significant improvements.

Summary of Manuscript:

Pelargonium graveolens L’Hèr. is valued for its traditional uses as a flavoring, ornamental, and medicinal plant, prompting an in-depth study of its phytochemical profile and antioxidant properties. UHPLC-HRMS analysis identified over 110 secondary metabolites, including acyltartaric and acylcitric acids, gallotannins, flavonoids, and phenolic acids, with acylcitric acids being reported in this species for the first time. The methanol-aqueous extract demonstrated significant antioxidant activity, scavenging DPPH and ABTS radicals, and exhibiting high reducing power in CUPRAC and FRAP assays. The extract showed metal chelating capacity and high performance in the phosphomolybdenum assay, indicating strong antioxidant potential. It also exhibited notable enzyme inhibition, affecting acetyl- and butyrylcholinesterase, tyrosinase, α-glucosidase, α-amylase, and lipase, suggesting potential applications in pharmaceutical and nutraceutical products.

Comments for authors are as follows:

1-      The figure 2 contents are too small unable/hard to read; I suggest authors to provide the figure 2 with large front size.

Response: Dear Reviewer, thanks for the comment. The Figure was corrected appropriately.

2-      Table 2: authors written “Total flavonoid content content (mg RE/g)” delete the repeated words.

Response: Dear Reviewer, thank you for your remark. The correction was done.

3-      Line 463-467: how many replications were used? How many number of leaves were t=collected for metabolic determination?

Response: Dear Reviewer, thank you for your remark. 100 g leaves were collected from seven plants and were subjected to a further extraction with 80% of methanol. Triplicate analyses were performed for the LC-HRMS. The used mzMine software did not show any significant differences in the dominant compound in each group. In full MS and MS/MS experiments the mass accuracy was within 5 ppm for different replicates. A text is embedded: “As the fresh/dried mass ratio of G. graveolens leaves is 1:6 (Nikolov, 2006), 600 g of fresh biomass of rose geranium leaves was used to obtain 100 g dry plant material.”

4-      Line 464-465: “The plant material (leaves) was dried in the laboratory for one week at room temperature (20-22°C) and 50% of relative humidity.” Why authors dry the leave at room temperature for one week??? Does this drying procedure may change the inter secondary metabolic profiles. Please provide a strong reference for using this method to dry the leaves at room temperature.

Response: Dear Reviewer, thank you for your remark. An explanation for the reason this drying procedure was accomplished is presented and strong references are added in the text: “The drying procedure at low temperature (20-22°) aims plant material does not remove water completely which would not interfere with further analysis due to change in surface activity of the stationary phase (Romanik et al., 2007). Moreover, the leaves were dried until a constant weight of the plant material was obtained (Djordjevic, 2017).”

5-      Line 470: “Air-dried powdered leaves (100 g) were extracted” Is it 100 g??? Why to high> maybe it is 100 mg; how many replications?

Response: Dear Reviewer, thank you for your remark. 100 g air-dried powdered leaves were extracted with 80% MeOH, than the organic solvent was evaporated and the water residue was lyophilized. Then, the lyophilized extract was used for further in vitro biological tests. In the in vitro tests around 100milligrams of lyophilized extract are needed for to be accurately weighed and triplicate analyses of all tests. Moreover, future in vivo tests will be performed, and they need a quantity of more than 2 grams for the investigations. For that reasons we used 100 g not 100 mg. 

6-       Line 470-476: where is reference?

Response: Dear Reviewer, thank you for your remark. Reference is embedded in the text: “Romanik et al., 2007”.

7-      Line 479: “The reference standards” write the standard name and concentration and amount of standard used…..

Response: The names of reference standards are written. The used concentrations were added (See 3.3. Chemicals).

8-      “3.5. Assay for total phenolic and flavonoid contents According to the methods specified by [73], total phenolics and flavonoids were quantified. Gallic acid (GA) and rutin (RE) served as standards in the assays, and the out- comes were reported as gallic acid equivalents (GAE) and rutin equivalents.” Write the full details of phenolic and flavonoid assay. How samples were extracted??? how many sample was weighted ?? What was the standard concentration?? What was the control?? Chemical used in these assay were purchased from which company??? Writing just reference is not enough….

Response: Appropriate references have been added to support the protocol used. For the sake of similarity, we have not included all details of the protocols as these have been previously published. However, all experimental details have been given in the supplemental materials. In addition, the chemicals used in the tests were included in the revised paper.

9-      Same questions “3.6. Assays for in vitro antioxidant capacity

10-   According to the methods provided by [74], antioxidant tests were executed. The DPPH, 506 ABTS radical scavenging, CUPRAC, and FRAP test results were expressed as milligrams of Trolox equivalents (TE) per gram of extract. The antioxidant potential determined by the phosphomolybdenum (PBD) assay was measured in millimoles of Trolox equivalents (TE) per gram of extract, and metal chelating activity (MCA) was conveyed as milligrams of disodium edetate equivalents (EDTAE) per gram of extract.” Write the full details of phenolic and flavonoid assay. How samples were extracted??? how many sample was weighted ?? What was the standard concentration?? What was the control?? Chemical used in these assay were purchased from which company??? Writing just reference is not enough….

11-   Same questions “”Write the full details of phenolic and flavonoid assay. How samples were extracted??? how many sample was weighted ?? What was the standard concentration?? What was the control?? Chemical used in these assay were purchased from which company??? Writing just reference is not enough….

Response: Dear Reviewer, thank you for your remarks. Appropriate references have been added to support the protocol used. For the sake of similarity, we have not included all details of the protocols as these have been previously published. However, all experimental details have been given in the supplemental materials. In addition, the chemicals used in the tests were included in the revised paper.

12-   Which statistical program was used to analysis the data; write the details in the methods section.

Response:  Thanks so much for your comment. We have added statistical subsection in the revised version.

13-   Move the table 1 to supplementary files.

Response: Dear Reviewer, thank you for your remark. The Table 1 was moved to the Supplementary material. Accordingly, a new Table 1 was created in the main documents where the annotated/dereplicated compounds were listed.

Reviewer 2 Report

Comments and Suggestions for Authors

The manuscript, entitled "Pelargonium graveolens: towards in-depth metabolite profiling, antioxidant and enzyme inhibitory potential," presents data on the chemical composition of geranium, including polyphenols, flavonoids and phenolic acids. The study also investigated their antioxidant activity (e.g. DPPH, FRAP, and iron chelation ability assas) and anti-enzyme activity. This article is interesting and a great contribution to the knowledge in this field. The introduction is clearly worded and the results are well presented. I suggest a few changes before being accepted for publication.

1. Introduction

- Please tidy up the footnotes. For example, 89-93 simply adds a footnote at the end of the paragraph

- In the introduction, the phytochemical composition should be given first, followed by a description of the biological characteristics.

2. Results and discussions

- The first time an abbreviation is used, the full name must be given, e.g. FRAP, CUPRAC

-347 Do the values in the table refer to a concentration or an IC50 value?

- In general - When reporting results from other investigators, information about enzyme concentration, incubation time, reaction pH, etc., should be included. Many factors can affect the outcome.

Step 3: Method

- 472 freeze drying and evaporation temperatures are required

- In general - methods for testing antioxidant, anti-enzyme properties, and TPC and TFC should be described in detail

In general, methods for testing antioxidant, anti-enzyme, and TPC and TFC properties should be described in detail

4. Conclusion

- The study is a pilot study and further points should be pointed out

Author Response

Reviewer 2

The manuscript, entitled "Pelargonium graveolens: towards in-depth metabolite profiling, antioxidant and enzyme inhibitory potential," presents data on the chemical composition of geranium, including polyphenols, flavonoids and phenolic acids. The study also investigated their antioxidant activity (e.g. DPPH, FRAP, and iron chelation ability assas) and anti-enzyme activity. This article is interesting and a great contribution to the knowledge in this field. The introduction is clearly worded and the results are well presented. I suggest a few changes before being accepted for publication.

  1. Introduction

- Please tidy up the footnotes. For example, 89-93 simply adds a footnote at the end of the paragraph

Response: Dear Reviewer, thank you for your remark. The changes were done.

- In the introduction, the phytochemical composition should be given first, followed by a description of the biological characteristics.

Response: Dear Reviewer, thank you for your remark. According to the Reviewer’s suggestions, the Introduction was modified: first, the phytochemical composition was given, followed by the pharmacological properties.

  1. Results and discussions

- The first time an abbreviation is used, the full name must be given, e.g. FRAP, CUPRAC

Response:  Dear Reviewer, thank you for your remark. We have explained the abbreviations in the first place.

-347 Do the values in the table refer to a concentration or an IC50 value?

Response: Dear Reviewer, thank you for your remark. The table was referred as standard equivalent values for each antioxidant and enzyme inhibition assays. We have used positive controls for all antioxidant and enzyme inhibitory assays. The antioxidant properties were expressed as standard compounds including trolox (for radical scavenging, reducing power and phosphomolybdenum) and EDTA (for metal chelating). The enzyme inhibitory activities of the extracts were evaluated as equivalents of standard inhibitors per gram of the plant extract (galantamine for acetylcholinesterase and butyrylcholinesterase, kojic acid for tyrosinase, and acarbose for α-amylase and α-glucosidase inhibition assays listed hereunder. In this perspective, our results tend to align with previous peer reviewed published scientific literature (Peron et al (2024) Journal of Functional Foods, 116, 106147; Elhawary et al (2024) BMC Complementary and Therapies, 24 (1), 73; Stefanucci et al (2024) Food Biosicence, 57, 103539).

- In general - When reporting results from other investigators, information about enzyme concentration, incubation time, reaction pH, etc., should be included. Many factors can affect the outcome.

Response: Thank you for your comment. We have added the enzyme concentration in the revised version.

Step 3: Method

- 472 freeze drying and evaporation temperatures are required

Response: Freeze drying and evaporation temperatures are added (See 3.2. Sample extraction).

- In general - methods for testing antioxidant, anti-enzyme properties, and TPC and TFC should be described in detail

Response: Dear Reviewer, thank you for your comment. Appropriate references have been added to support the protocol used. For the sake of similarity, we have not included all details of the protocols as these have been previously published. However, all experimental details have been given in the supplemental materials.

  1. Conclusion

- The study is a pilot study and further points should be pointed out

Response: Dear Reviewer, thank you for your comment. The further points were added (See Conclusion).

Reviewer 3 Report

Comments and Suggestions for Authors

I think that this paper is suitable to the Plants, and should be accepted.

Author Response

Reviewer 3

Comments and Suggestions for Authors

I think that this paper is suitable to the Plants, and should be accepted.

Response: Dear Reviewer, thanks for the comment.

Round 2

Reviewer 1 Report

Comments and Suggestions for Authors

The authors sufficiently revised the manuscript! Still, I have some suggestions!

I suggest authors include the protocols of antioxidant activities in the main manuscript! move from supplementary files to the main text.

The right abbreviation is "ABTS•+" and it is "radical cation (ABTS•+) scavenging assay" Please correct this throughout the manuscript.

Author Response

Reviewer 1

The authors sufficiently revised the manuscript! Still, I have some suggestions!

I suggest authors include the protocols of antioxidant activities in the main manuscript! move from supplementary files to the main text.

The right abbreviation is "ABTS•+" and it is "radical cation (ABTS•+) scavenging assay" Please correct this throughout the manuscript.

Response: Thank you for your comments. We have moved antioxidant methods in the main text ad corrected the name of ABTS•+.
